# Cultural Routes as Cultural Tourism Products for Heritage Conservation and Regional Development: A Systematic Review

**Xinyue Lin** [1,2]**, Zhenjiang Shen** [3,]*****[ID]**, Xiao Teng** [1,2] **and Qizhi Mao** [4]

[1] Graduate School of Environment Design, Kanazawa University, Kanazawa 920-1192, Japan; xinylin53@gmail.com (X.L.); riendyteng@staff.kanazawa-u.ac.jp (X.T.)
[2] Cross-Strait Tsinghua Research Institute, Xiamen 361006, China
[3] International Joint Laboratory of Spatial Planning and Sustainable Development (FZUKU-LAB SPSD), Fuzhou University-Kanazawa University, Fuzhou 350108, China
[4] School of Architecture, Tsinghua University, Beijing 100084, China; qizhi@mail.tsinghua.edu.cn
***** Correspondence: shenzhe@se.kanazawa-u.ac.jp

**Abstract:** Cultural routes are a composite set of heritage sites that refer to historical routes of human communication. As key products of cultural tourism, they provide visitors with rich cultural experiences across regions. We systematically review reports and studies related to the tourism development of 38 cultural route cases worldwide, with a special focus on their distribution, typology, planning patterns, and tools for cultural tourism. We summarized eight tools and found some differences in how often these eight tools are used by the different types of routes and different planning patterns for route tourism. This study also developed an evaluation system based on the conservation principles of cultural routes to determine how different tourism tools affect the conservation and development of historical regions. Although tourism decision-makers have made numerous efforts to protect and develop cultural routes, there are still many problems and challenges in the process of tourism development along cultural routes. We conclude the paper by making recommendations for decision-makers and researchers concerning future route tourism planning and study.

**Keywords:** cultural routes; tourism development; conservation principles; regional planning; tourism impact

## 1. Introduction

In recent years, cultural routes have been increasingly discussed as important products of cultural tourism that can bring tourists experiences across time and space. Previous research has described the contribution of cultural routes to sustainable tourism and socio-economic development through numerous cultural tourism cases [1–4]. However, several problems and challenges remain in establishing tourism planning for cross-regional cultural routes from the whole to the local level and how to create a systematic tourism development methodology and evaluation system. This study aims to answer these questions in terms of the current situation, types, models, and tools of cultural tourism development of cultural routes.

### 1.1. Definition of Cultural Routes

The cultural route is a relatively new concept in heritage conservation. It was first proposed in the 1980s by the European Commission in the revival of the Camino de Santiago pilgrimage route in Spain. In 1998, in order to achieve an international consensus on the concept of cultural routes, a minority of members of the International Council on Monuments and Sites (ICOMOS) established the International Committee on Cultural Routes (CIIC), which promulgated the ICOMOS Charter on Cultural Routes in 2008.

The charter provides a clear definition: Cultural routes refer to any historical routes of human communication having their own specific dynamic and historic functionality to

serve a specific and well-determined purpose [5]. Under the appeal of this international charter, various regions and countries have begun to pay attention to the protection and development of this large-scale heritage.

Cultural routes provide a way to interpret history and culture as a whole [6]. The process of developing cultural routes helps to discover various ancient routes with important historical and cultural significance and designates tangible and intangible resources along the routes as proof of their existence [7]. It can bring a new dimension to cultural heritage management through cross-regional cooperation [5], and can also be used as a tool to achieve local cultural cohesion and sustainable development [3,4].

### 1.2. Cultural Tourism and Heritage Sites

In 1995, the World Tourism Organization (WTO) introduced "cultural tourism" as a type of tourism that responds to consumers' own cultural motivations and needs (people's activities satisfy the need for diversity inherent in the human nature and tend to raise the cultural level of humankind by providing the opportunity for new knowledge, experiments, and meetings) [8]. The Committee on Tourism and Competitiveness (CTC) provided a clear definition of "cultural tourism" at the 22nd WTO General Assembly in 2018: Cultural tourism is a type of tourism in which the main goal of the traveler is to learn about, discover, experience, and enjoy the tangible and intangible cultural attractions and goods of the location. These attractions/products are unique to society and include art and architecture, historical and cultural heritage, gastronomic heritage, literature, music, creative industries, and living cultures with their ways of life, value systems, beliefs, and traditions [9].

Cultural heritage sites with long histories are the most significant cultural tourist attractions. The tourist industry and the accompanying protection and interpretation of heritage sites are key instruments for the sustainable development of heritage sites [10,11], helping to build territorial cohesion and strengthen local identity while improving the local economy [5,12,13]. However, over-tourism also has the potential to cause problems, such as the destruction of cultural artifacts, environmental pollution, and loss of residents in heritage communities [10,14].

### 1.3. Cultural Routes as Cultural Tourism Products

Cultural routes are intertwined with cultural tourism. Cultural routes can bring together numerous tourist attractions in the region (some of which may be less well known) to market them more effectively under a unifying theme and improve the management and conservation of heritage assets. Tourists can create a comprehensive experience of local history and culture through various heritage landscapes while traveling along the routes, thus achieving the goal of cultural tourism [2,15]. Simultaneously, there being several different heritage sites linked along the route makes the preservation of cultural routes more difficult and complex, requiring new instruments for the assessment of their protection and conservation. For this purpose, ICOMOS proposed that the protection and promotion of a cultural route can harmoniously integrate with tourist activities [5]. The Council of Europe also stated that promoting cultural tourism is a natural next step in creating cultural routes because this type of tourism builds on the uniqueness and genuineness of distant locations, local knowledge, skills, history, and customs [16].

A growing number of cultural tourism initiatives are linked to cultural routes. This study presents a systematic review of the existing academic and professional literature on cultural tourism development along cultural routes. We sought to summarize the status and tools of cultural tourism development and their impact on the conservation of cultural routes and regional development. Our results are intended to facilitate the cross-fertilization of tourism and cultural route conservation experiences and inspire and provide recommendations for the planning and management of cultural routes.

## 2. Methodology

This review can be divided into 3 steps: (1) selection of studies to be included, (2) analysis and classification of cases, and (3) summarization and evaluation of tools for tourism development along the routes.

### 2.1. Selection of Studies to Be Included

As shown in Figure 1. First, we identified a comprehensive set of keywords. As "cultural routes" is a complex concept, a preliminary literature review found other two variations. We present an overview of these concepts in Table 1. These keywords were used to identify the relevant studies cited in the Web of Science (core collection) and Google Scholar (contains database content from Scopus, IEEE Xplore, ScienceDirect, and other publishers and organizations). The searches were not limited by language but were limited to studies published after 2008, when ICOMOS published the ICOMOS Charter on Cultural Routes. A total of 3498 references were identified after removing duplicates. These studies had diverse content, and some did not precisely meet the objectives of our study. Therefore, by including the qualifiers "heritage conservation" and "cultural tourism" in further screening, the number of references was reduced to 900.

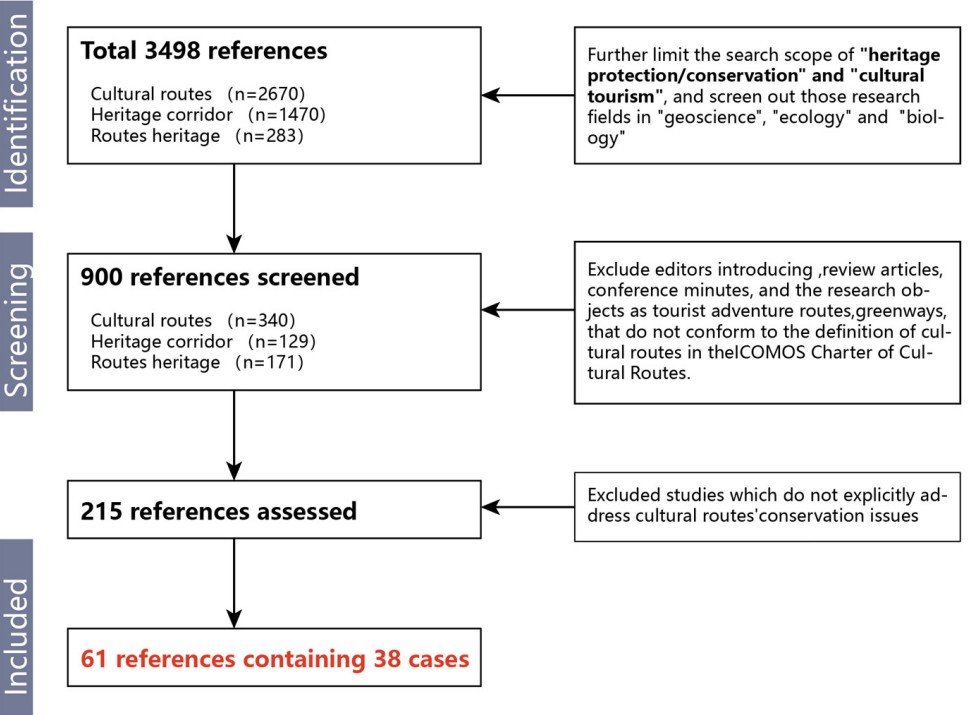

**Figure 1.** The process of selecting cases for systematic review.

Based on the title and publication information, we excluded the editor introductions, review articles, and meeting speech minutes. In addition, some studies considered routes that do not matching up with the characteristics of cultural routes (Table 2), such as tourist routes and greenways. To avoid conceptual confusion, such studies were excluded through abstracts and keywords, and the number of studies was reduced to 215. Literature that did not substantively address the problem of protecting and managing cultural routes was eliminated through full text browsing. Finally, 61 references were selected for this study, including 38 cases of cultural routes. The case and the references can be found in the Appendix A.

**Table 1.** Identification of concepts used as keywords.

| Keywords | Definitions |
|---|---|
| Cultural route | Cultural routes display route systems of cultural assets and historical sites created by cultural exchange and dialogue. These routes can integrate spiritual, economic, environmental, and cultural values into tourism systems [17]. |
| Heritage corridor | Cultural corridors are networks of cultural creativity and economic exchange based on a wide range of stakeholders. They form historical axes of ancient cultural and economic ties [6]. |
| Route heritage | A heritage route is composed of tangible elements of which the cultural significance comes from exchanges and a multi-dimensional dialogue across countries or regions and that illustrate the interaction of movement along the route in space and time [18]. |

**Table 2.** The overview of the characteristics of cultural routes.

| Keywords | Definitions |
|---|---|
| Origin | A geographically defined pathway of human movement that might have been created as a planned project or taken advantage (fully or partially) of pre-existing roads and evolved over a long period to fulfill a collective purpose [4]. |
| Context | Within a given cultural region or extended across different geographical areas that share a process of reciprocal influences in the formation or evolution of cultural values [5]. |
| Fundamental features | Long-lasting history with continuity in space and time, multi-dimensional function, wholeness, crossing and connecting borders, reflecting cross-fertilization of cultures (include a dynamic factor), associational value [3,5,19]. |
| Content | The communication routes itself, tangible heritage assets and intangible heritage elements [5]. |

## 2.2. Analysis and Classification of Cultural Route Cases

Second, we categorized the cases according to the region, function, context, and content of the cases, which helps understand the division of cultural route projects that have been identified and developed for tourist purposes, as well as the relevant institutions and policy supports in different countries. After classifying the routes, we classified tourism models according to the level of development and also analyzed the value that different models bring.

## 2.3. Summarizing and Evaluation of Tools for Cultural Tourism

We summarized the tools of tourism along cultural routes, which helped create a toolbox for tourism decision-makers matching different types of networks. Cultural tourism is a double-edged sword in heritage conservation. Tourism brings changes to the natural environment, transportation, and residents' lives around the route, thus affecting the authenticity and integrity of the heritage and the sustainability of its preservation and development. Therefore, after understanding different methods of tourism development through a literature review, it is necessary to establish an evaluation framework based on the conservation principles of cultural routes to determine the impact of different methods on route protection and promotion. Table 3 shows three principles that planners and heritage protectors pay more attention to when cultural routes are developed as tourism products.

**Table 3.** Evaluation framework based on the conservation principles of cultural routes.

| The Conservation Principle of Cultural Routes | Description | | References |
|---|---|---|---|
| Authenticity | Historical and cultural characteristics | Authenticity should be evident in the natural and cultural context to prove its historic functionality. | [5] |
| | Heritage discovery and restoration | Authenticity should be reflected in every part of the route and evident by the tangible and intangible heritage. | |
| Integrity | Historical and cultural background | To ensure that the significance of the cultural and historical processes of routes can be fully demonstrated. | [5] |
| | Route structure | Evidence of the historic relationships and dynamic functions essential to the distinctive character of the cultural route, whether its physical fabric and/or its significant features are in good condition, and whether the impact of deterioration processes is controlled. | |
| Sustainability | Environmental protection | To protect the environment and natural landscape around the routes. | [2] |
| | Activation and utilization | To promote local community well-being and economic development. | [2,20] |
| | Education and promotion | To improve the visibility of the routes and people's awareness of heritage conservation. | [21] |

## 3. Distribution and Classification of Cultural Route Products

### 3.1. Case Distribution

Of the 38 cases covered in the literature (some of which were overlapping in parts), the distribution was mostly in the USA (n = 6), China (n = 6), and European countries (n = 19), with several cases spanning multiple countries (Figure 2). This suggests that cultural routes have been well studied in these three regions, and that there are political systems or institutions for heritage management and tourism development to support the conservation and utilization of such composite heritage across regions.

Through the National Heritage Area (NHA), the United States has established a series of mature, interfederal programs that combine natural, cultural, and historical resources. In Europe, the European Commission (COE) established the Institute for Cultural Routes (EICR) in 1998. As an important part of the new cultural heritage policy in Europe, the Cultural Routes Programme combines cooperation in the fields of literature, educational heritage, and tourism. In China, the Guidelines for the Protection of Cultural Relics and Monuments (2015) also include cultural landscapes, cultural routes, and heritage canals as new heritage types in cultural heritage protection systems.

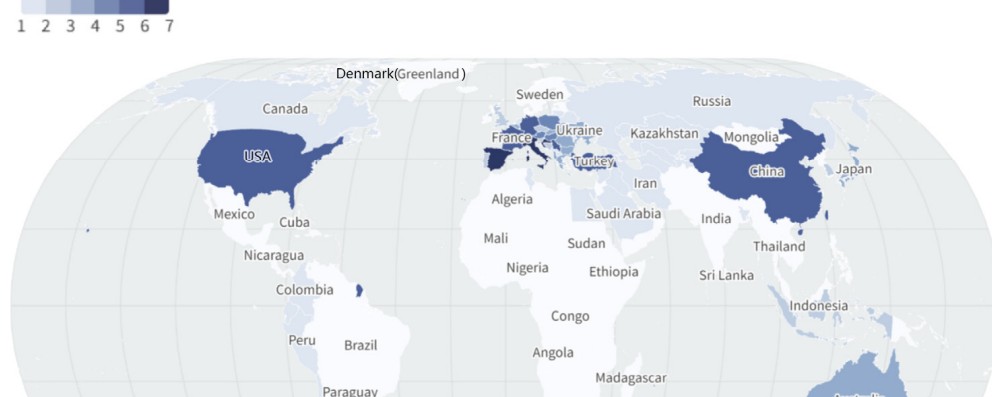

**Figure 2.** Case distribution.

*3.2. Case Classification*

The origins of cultural routes are diverse. We categorized these networks into three main categories and seven sub-categories based on their functions, contexts and contents (Table 4).

**Table 4.** Route classification according to their functions, context, and content.

| Categories | Description | Subcategories | References |
|---|---|---|---|
| A | Routes used for specific events in a period of history | Pilgrimage route | [18] |
| | | Military route | |
| B | Routes defined with the use of heritage or landscape | Heritage canal or valley | [22] |
| | | Historical border | |
| | | Railway heritage | |
| | | Highway heritage | |
| C | Important routes in ancient times because of the need for transportation and trade | Trade, migration, or transportation routes | [5,22] |

As it is shown in Figure 3, of the 38 cases covered by the literature, Category B (heritage canal or valley, historical border, railway heritage, and highway heritage) has the highest number of tourism projects associated with the routes (n = 16) because they are based on nature (canals or valleys) or large heritage (e.g., ancient city walls or railway tracks) itself, and the remains of their linear mechanisms can be better excavated and preserved. Category A (pilgrimage and military routes), although fewer in number (n = 9), is generally better preserved due to the specific historical and cultural contexts. There is a series of governmental institutions, academic organizations, and civil society groups related to the preservation of these networks, particularly pilgrimage routes in both Japan and Europe [7,23,24]. Conversely, Category C (ancient trade, migration, or transportation routes) often crosses multiple regions or even countries (e.g., the Silk Road and the Phoenician Way). With changes in urban development and transportation modes, these routes have lost their own functional significance, and the relics around the routes have suffered serious damage, making them relatively difficult to protect, excavate, and develop for tourism (n = 13).

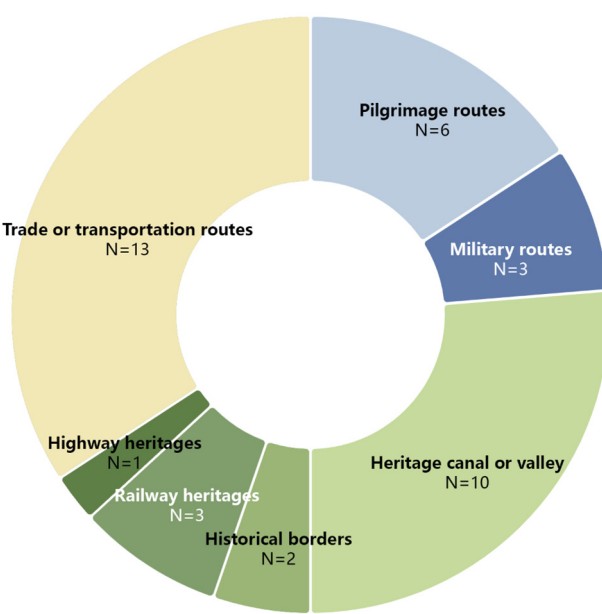

**Figure 3.** Different types of cases to be included (N = number of cases of this type).

## 4. Planning Patterns for Route Tourism

Planning can be considered a set of actions, choices, and decisions that are made systematically to reach a set of goals. This is done by considering all existing strengths, weaknesses, opportunities, threats, policies, facilities, and challenges. Cultural route planning is an area of tourism and conservation planning that has grown over the last few decades [3,24]. There are several different meanings and classifications of cultural routes; however, there is no clear system for planning cultural routes, and there is no general way to handle them. This study found that cultural routes have brought new planning approaches to cultural tourism, which we documented and summarized into the following three route tourism planning patterns: point-based pattern, linear-based pattern, and area-based pattern (Figure 4).

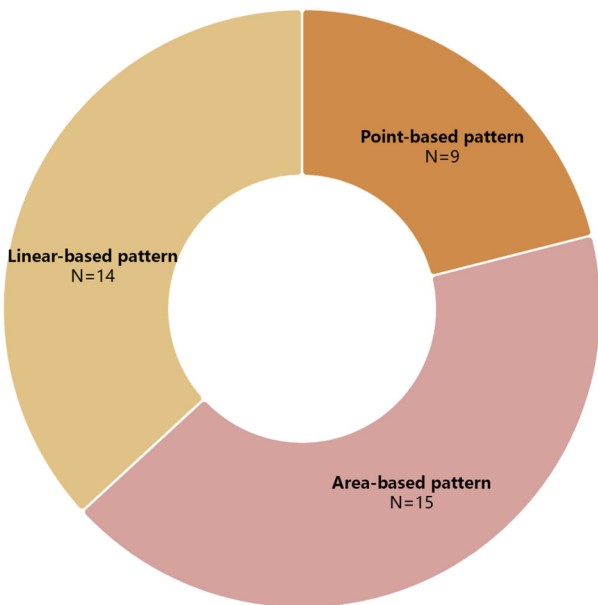

**Figure 4.** Different planning patterns for route tourism (N = number of cases of this pattern).

*4.1. Point-Based Pattern*

In this pattern, the planning of the route consists of a series of cultural heritage sites that present different scales and are linked by a theme; in other words, the routes lack territorial continuity and provide only intangible connections. For example, the Silk Road consists of a series of monuments spanning Europe and Asia [25–27] (Figure 5), which do not represent the same historical/cultural group or a uniform type of heritage within the agreed-upon geographical area [28]. This is because various socio-political factors such as regional conflicts and changing markets meant that the Silk Road was not static and consisted of a wide variety of paths, tracks, and roads. Many of the cities along the route have grown to become major cultural and artistic centers, deriving power and wealth from trade, providing infrastructure for production and redistribution, and regulating routes within their borders [29]. Today, as transport modes have evolved, these routes no longer exist, but many remnants of the intermingling of different ethnic and cultural backgrounds remain in the cities. These sites are now landmarks for travelers exploring the traces of the Silk Road, and the passage of this ancient trade route has added value and symbolism to these sites [25,29].

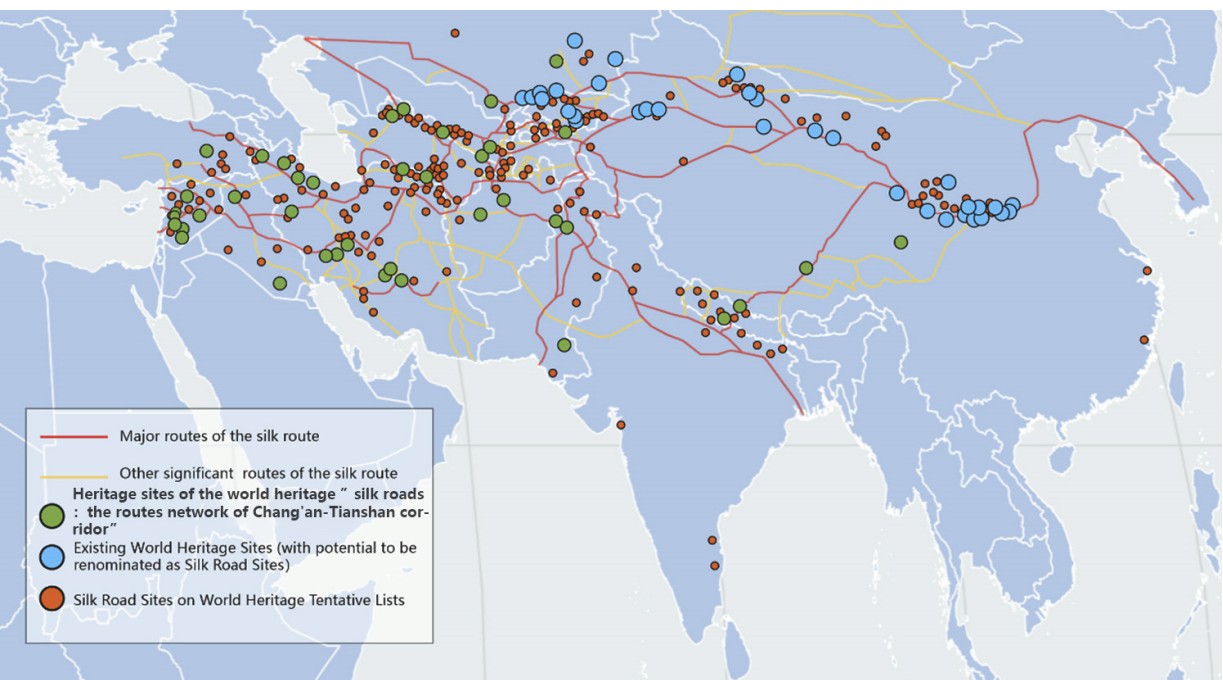

**Figure 5.** Silk Road and heritage locations for World Heritage. Source: [30] and "https://www.silkroadfutures.net/silk-road-world-heritage-page (accessed on 21 July 2023)".

*4.2. Linear-Based Pattern*

In this pattern, the main element constituting the route is usually the path itself. As in the case of marching routes and pilgrimage routes, these routes are special and unique and hardly change over time. Routes have evolved, transformed, and strengthened over time, establishing close connections with the territories they traverse. As routes are part of regional territories, these places themselves are transformed by the routes and the travelers, which enable these territories to be culturally connected through the routes even if they are geographically far apart. For example, the masterplan of the pilgrimage route from Sarria spans about 100 km (it is part of the Camino de Santiago pilgrimage route), with several municipalities along the official route collaborating to plan and implement projects to produce uniform wayfinding signage, street furniture, restoration works, key stops, and rest areas along the route, among others, to provide a safe travel plan for pilgrims on foot or by bicycle [31,32] (Figure 6). It takes several days for visitors to experience the entire

route in its entirety, and as they travel, they encounter ancient transport infrastructure, pilgrim shelters, and towns that have developed because of the route [32,33].

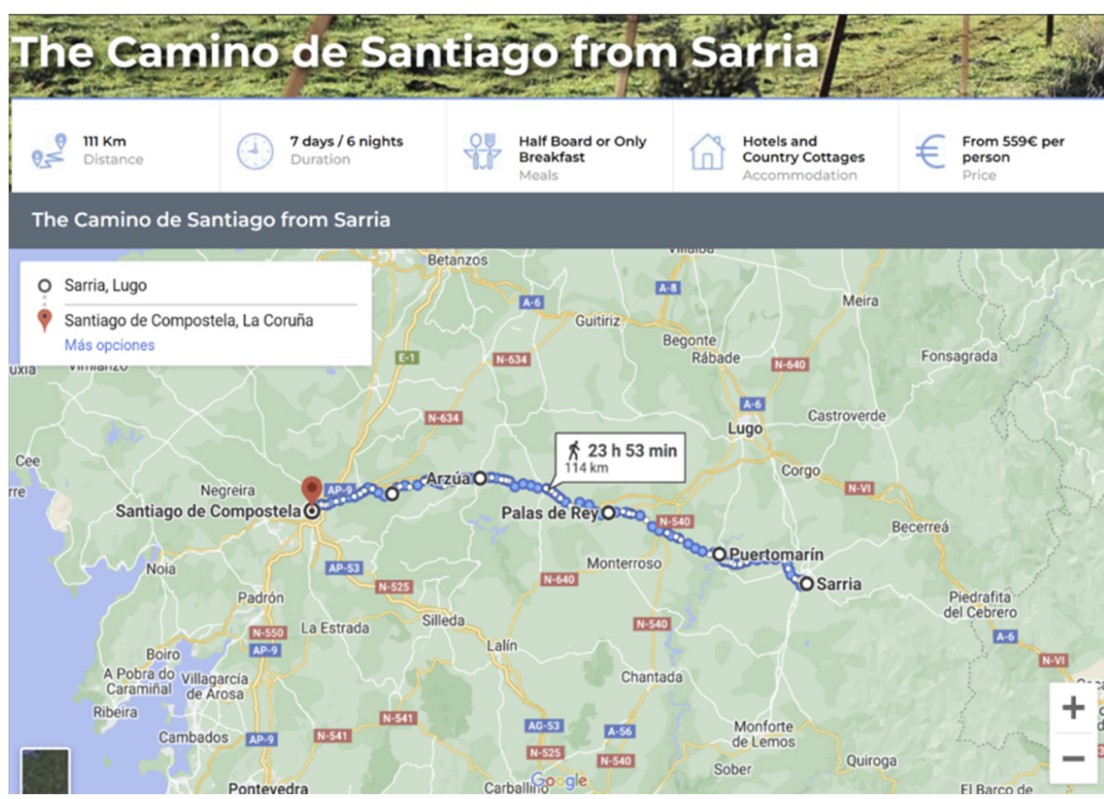

**Figure 6.** Map of the Camino de Santiago from Sarria and tourism services. Source: "https://santiagoways.com/en/camino-de-santiago-routes/camino-frances/the-way-of-saint-james-from-sarria/ (accessed on 20 July 2023)".

### 4.3. Area-Based Pattern

The main elements comprising the routes in this pattern are themes and natural landscapes that are shared on a regional scale, such as canals, mountains, and farms. Routes such as shipping, commerce, and migration develop themes of civilizational currents, imbue cross-regional natural landscapes with culture and identity, and, based on this theme, continually link culturally distant but geographically adjacent regions. For example, the Blackstone River Valley Heritage Corridor in the United States includes mill towns stretching across 25 cities and towns near the river's course in Worcester County [34] (Figure 7). There are multiple historic districts within the conservation area, and visitors can start anywhere and choose single or multiple historic districts to experience the industrial landscapes, agricultural landscapes, national monuments, etc., along the canal. Regardless of where it began, the cultural landscape, including mill villages, rural villages, monuments, transportation systems, and waterways integrated into an industrial network, speaks to the history of the thriving American industry of the 18th century [35,36].

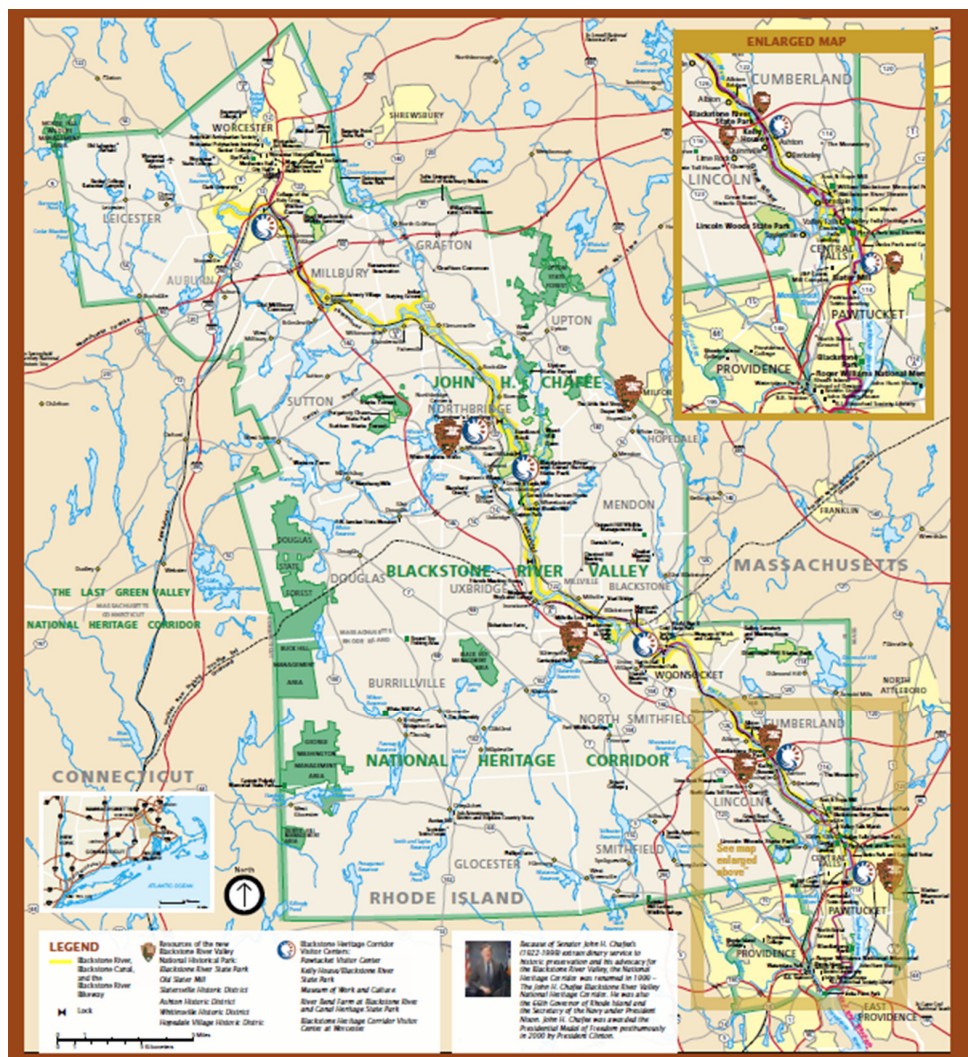

**Figure 7.** The Blackstone River Valley heritage corridor and tourism regions. Source: "https://blackstoneheritagecorridor.org/exploring-the-blackstone-river-valley/maps-tours-guides/ (22 July 2023)".

## 5. Tools of Cultural Tourism for Regional Conservation and Development

### 5.1. The Summary of Tools

After a systematic literature analysis, we summarized eight types of tools for the tourism development of the cultural routes mentioned in 61 references (Figure 8). In almost all case studies (n = 35), route committees or associations were considered by researchers to be a tool bring together socioeconomic and cultural actors and agents interested in and affecting the governance and dynamics of the route to design a more rational tourism plan. It has particularly come into discussion for routes that cross different countries or regions (e.g., the Danube region [37] crosses 14 countries in Europe, and the Silk Road crosses 33 countries in Europe and Asia [25]) that involve a wide range of stakeholders (e.g., the preservation of the Way of St. James involves residents, church clergy, city planners, tourists, tourism industry players, etc. [38]).

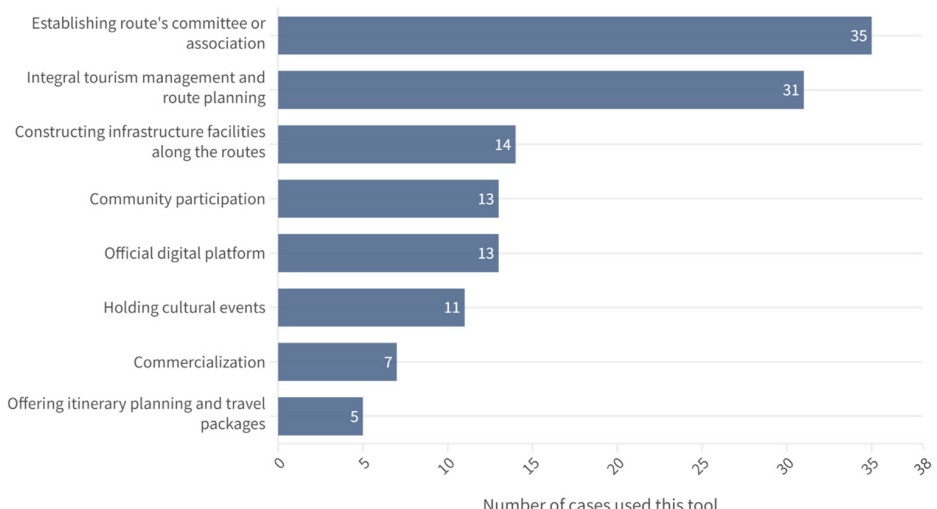

**Figure 8.** Tools of cultural tourism for regional conservation and development.

The second most common type of study conducted integral tourism management and route planning (n = 31); however, it is difficult to adopt large-scale routes with poorly preserved historical relics. For example, the Silk Road and the Phoenicians' Route have lost most of the specific traces of their routes, and only important nodes can be traced.

Moreover, according to the literature review, to make the routes more attractive and increase the accessibility of important sites along the routes, the route committee establishes a series of transport and artistic infrastructure along the cases, such as museums, art galleries, highways, and stations (n = 14); organizes regular cultural events (n = 11); creates official information platforms for tourists to access the history and attractions along the routes; updates the information about the events regularly (n = 13); and appropriately commercializes some of the historical areas (n = 7). Some cases provide multi-day travel packages with accommodations, food, and guided tours (n = 5). To make tourism development more sustainable, communities along the route were sometimes called upon to participate in tourism development and route planning from the bottom up (n = 13).

From the analysis of the information gathered, it is clear that there are some disparities in the frequency of usage of these eight tools for the different types of cultural routes and different planning patterns for route tourism (Figures 9 and 10).

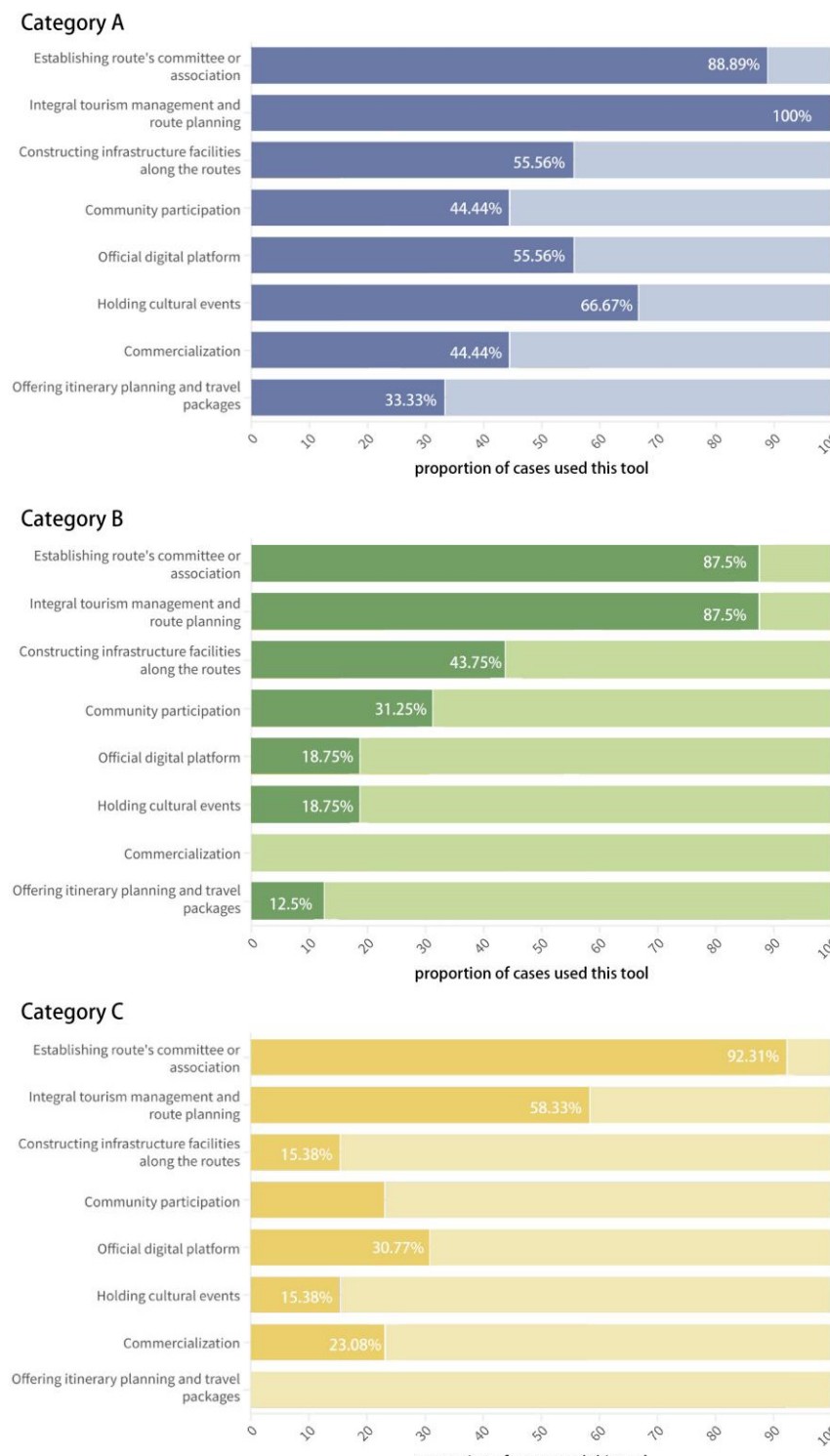

**Figure 9.** Different types of cultural routes and proportion of usage tools.

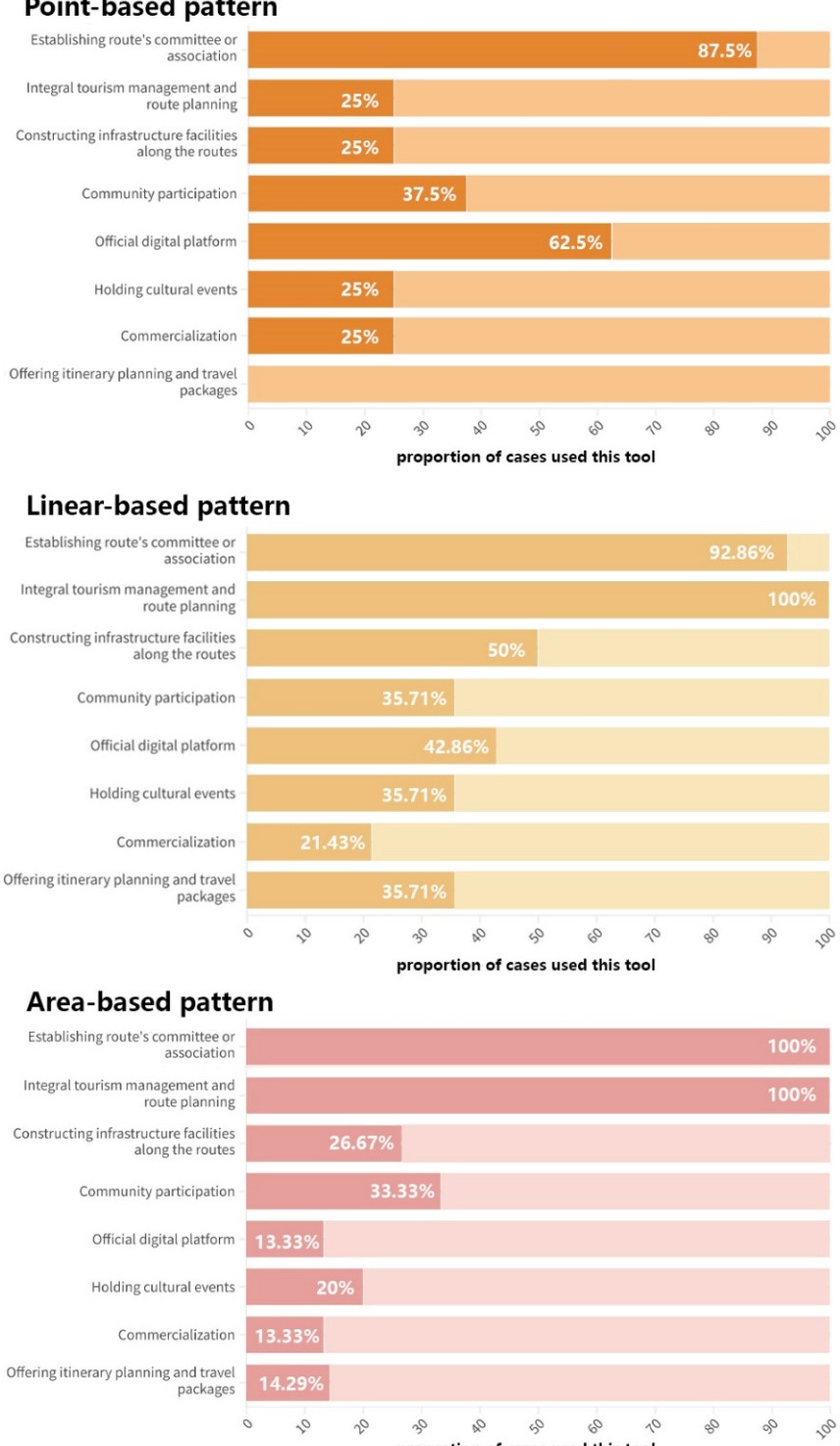

**Figure 10.** Different planning patterns for route tourism and proportion of usage tools.

*5.2. The Evaluation of Tools*

This study created an assessment framework based on the conservation principles of cultural routes and evaluated the eight tourism development tools mentioned in Table 5. The results show that cultural tourism contributes to the authenticity, integrity, and sustainability of cultural routes, but that there are some problems and challenges. The benefits and disadvantages of each tool are discussed in detail below.

**Table 5.** Tool evaluation.

| Tools of Cultural Tourism | Authenticity | | Integrity | | Sustainability | | |
|---|---|---|---|---|---|---|---|
| | Historical and Cultural Characteristics | Heritage Discovery and Restoration | Historical and Cultural Background | Route Structure | Environmental Protection | Activation and Utilization | Education and Promotion |
| Establishing route committees or associations | √ | √ | √ | √ | √ | - | √ |
| Integral tourism management and route planning | √ | √ | √ | √ | √ | √ | / |
| Constructing infrastructure facilities along the routes | √ | √ | √ | √ | / | √ | / |
| Community participation | √ | / | / | / | √ | √ | √ |
| Establishing an official digital platform | √ | / | √ | / | √ | / | √ |
| Holding cultural events | √ | / | √ | / | × | - | √ |
| Commercialization | × | - | / | / | × | √ | / |
| Offering itinerary planning and travel programs | √ | / | √ | / | / | / | √ |

√: Positive; ×: negative; -: ambiguous; /: uncorrelated.

### 5.2.1. Establishing Route Committees or Associations

The establishment of route committees or associations creates a platform that facilitates cooperation and communication among various stakeholders (e.g., governments, NGOs, residents, tourists, experts in various disciplines, etc.), which contributes to heritage management, conservation and revitalization, financing and management, itinerary design, and route promotion and publicity [39–43]. However, tourism may introduce too many stakeholders, which can affect the progress of heritage conservation and development [41].

In the case of routes that cross several countries or even continents (e.g., the Phoenician Way, the Roman Emperors and Danube Wine Route, the Silk Road, the Qhapaq Ñan, the Danube region), this approach can help countries to cooperate, communicate, and discuss with each other, and strengthen cooperation with international organizations such as the United Nations Educational, Scientific and Cultural Organization (UNESCO) and the World Tourism Organization (WTO) [3,25,44–49].

In addition to contributing to the conservation of individual routes, this approach facilitates cooperation between routes. For example, the committees of Kumano Kodo in Japan and the Camino de Santiago in Europe have established a close partnership. Santiago de Compostela and Tanabe have become twin cities and have granted "dual pilgrim status" to visitors who complete pilgrimages along the two routes. They work together to find innovative ways of harnessing cultural, religious, and natural heritage along pilgrimage routes [23].

### 5.2.2. Integral Tourism Management and Route Planning

Integral tourism management and route planning includes comprehensive onsite surveys and interviews of heritage sites, collection of historical documents, identification of routes on old maps, and restoration of ancient routes [50]. Sustainable tourism development assessment is built on comprehensive heritage research [37,51], and several places have also established a multi-level management and development system from the whole to the local level [37,39,52].

By these means, travel itineraries connect the unconnected and unknown but important archaeological sites along the route [44,53,54], allowing travelers to focus more on

the travel process than on the destination [55]. This contributes to the overall research, preservation, and revitalization of tangible and intangible cultural heritage along the route, and promotes the average development of each area on the route.

### 5.2.3. Constructing Infrastructure Facilities along the Routes

The construction of infrastructure facilities along the routes includes transportation and cultural infrastructure. The establishment of transportation infrastructure includes the restoration of ancient paths to connect important archaeological sites [50,56,57], constructing stations and highways to increase the accessibility of the route [57,58], and adding bicycle paths or jogging paths along ancient routes as part of thematic tourism activities to increase the interest in the routes' tourism [41,59,60].

As for cultural infrastructure, routes' museums and art galleries are good places to display the history and culture of the region so that tourists can establish a comprehensive understanding of the historical background and cultural characteristics of the sites [43,57,61–65]. For example, more than 60 art venues have been established along the Erie Canal in an attempt to make the canal an outdoor "art gallery," which has become a new landmark for tourism where tourists can enjoy art museums and galleries, locally based artistic traditions, and historical theaters [64].

### 5.2.4. Community Participation

The development of cultural tourism has changed the lives of regional communities. Accordingly, some places along the route seek to develop community-oriented and appropriate ways to preserve and utilize heritage sites. Community involvement raises local awareness of cultural and heritage preservation and increases community integration and people's sense of place [37,49,66]. Simultaneously, inhabitants are more focused on respecting local cultural heritage than attracting large numbers of tourists to increase economic benefits, which also contributes to more sustainable tourism development [23].

### 5.2.5. Establishing an Official Digital Platform

In recent years, several cultural routes have implemented actions to digitalize the tourism sector to improve communication before, during, and after the trip, achieving intensive and intelligent use of a digital technology ecosystem. The approaches include the Internet of Things, augmented reality and virtual reality, artificial intelligence and robotization, geolocalization and TIG, big data, small data, and open data. Some routes, such as the Camino de Santiago, have official websites where historical, cultural, and tourist information about the route is visualized and information about activities along the route is regularly updated (Figure 11) [2,45,59,67,68]. These sites enrich tourists' knowledge of the history and culture of the routes and help them to plan their respective itineraries based on an overall understanding of the region.

Additionally, digital technologies are used in the restoration and preservation of tangible and intangible heritage along routes, educating tourists about history and culture, improving the professional capacity of heritage conservationists, monitoring the environment of sites, and managing the impact of tourism on sites (Figure 12) [66,68].

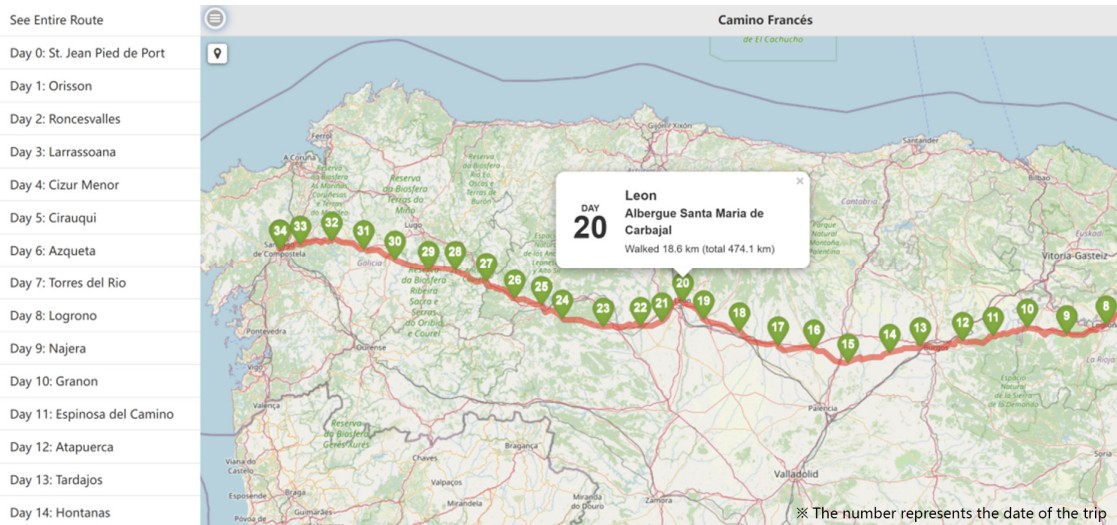

**Figure 11.** The interactive map on the website of the Camino de Santiago. Source: "https://caminofrances.info/ (accesssed on 23 July 2023)".

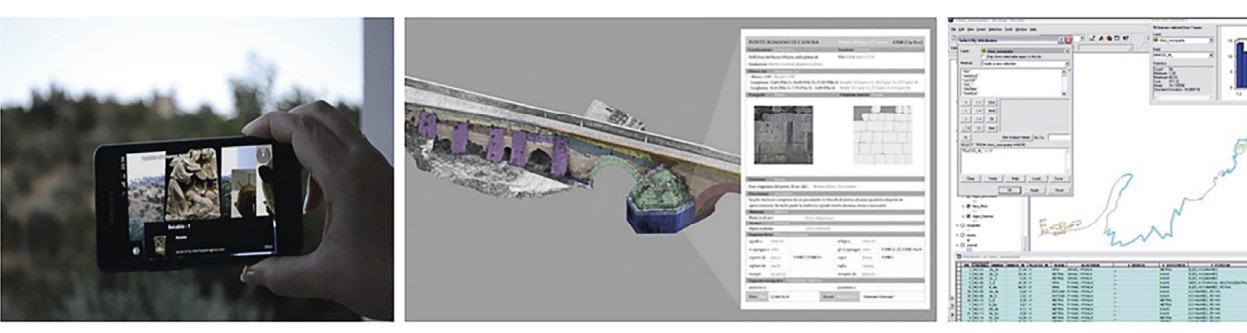

| The Museum AR application | 3D Reconstruction of a bridge along via Francigena | Statistical results in current preservation state produced by GIS management and analysis tools |

**Figure 12.** The digital technologies used in different aspects of cultural route protection and development. Source: [69–71].

### 5.2.6. Holding Cultural Events

Cultural and artistic events, celebrations, festivals, and ceremonies are often organized along cultural routes. Some of these events originate from traditional festivals [61], while others are organized to promote cultural tourism [39,63]. These activities stimulate tourists' interest during the trips, but they also carry the risk of overtourism, destroying the route's monuments, and polluting the environment [39].

Additionally, several routes have regular multi-expert seminars to present and discuss the findings of route research and achievements in heritage preservation. These acts encourage the multidisciplinary preservation of the routes' historical and cultural context [50].

### 5.2.7. Commercialization

Along cultural routes, there will always be an increase in the number of commercial activities as a result of cultural tourism, such as tourism marketing [39,49,72], craft selling [73], commercialization of historical buildings, and other similar endeavors [51,74]. Commercialization can help raise funds for heritage restoration and promote economic development in less developed areas along the route [51]; however, it can also bring about overtourism, resulting in environmental pollution and damage to monuments in historical area [51,74]. Moreover, the sale of handicrafts that differ from local traditions and the

construction of commercial buildings that differ from local architectural styles can threaten traditional ways of life and lead to the secularization of symbolism in many parts of the route [51,73,74].

5.2.8. Offering Itinerary Planning and Travel Programs

As cultural routes are generally large in scale, making it difficult for tourists to experience several attractions in a single day, various route committees offer multi-day route travel packages that even include meals, lodging, and tour guide services [31,62,75,76]. For example, the Path of Peace in Italy offers packages to visitors and school groups for educational purposes. The package of the Great War Anniversary trip comprises a stay of two to four days, visits to strongholds and trenches, as well as historical lectures given at the locations visited [5]. It greatly enriches the travel experience for tourists and strengthens the historical and cultural education and publicity of the route.

**6. Discussion**

*6.1. How Cultural Routes, as Cultural Tourism Products, Can Contribute to Heritage Conservation and Regional Development*

As a cultural tourism product, cultural routes contribute significantly to regional heritage conservation development. First, the increase in tourism activity has brought new employment opportunities and sources of income, which include the sale of local handicrafts, tourism services, and related catering and accommodations industries. In this way, it not only strengthens the economic base of the region but also provides local governments with the funds needed to restore and develop cultural heritage. In addition, cultural routes as part of cultural tourism can also help to raise public awareness of the importance of historical and cultural heritage. By demonstrating the value of heritage to visitors, it can foster public awareness of conservation, thereby facilitating wider community participation and input. The cultural routes can also promote cultural exchange and cooperation across regions and disciplines, which not only helps to share best practices and experiences for more effective conservation and use of cultural heritage but also enhances respect and understanding of each other's culture within and outside the region.

*6.2. How to Promote the Planning and Management of Cultural Routes*

In order to facilitate the planning and management of cultural routes and maintain the balance between tourism development and heritage conservation, several key steps can be taken to improve efficiency and sustainability.

(1) To establish a comprehensive planning and management framework. This should include detailed planning from the regional to the local level, ensuring that the needs and expectations of all relevant stakeholders are taken into account.

(2) To implement performance assessment tools to integrate goals related to specific interests of heritage conservation, local aesthetics, recreational activities, and various stakeholders. Such tools should enable effective assessment at different stages of tourism development to help balance the preservation and use of cultural heritage while avoiding the secularization of the symbolic meaning of cultural routes.

(3) To select the appropriate planning pattern. Depending on the specific content and function of the cultural route, point-, linear-, or area-based planning patterns can be selected. These patterns should be flexibly adapted to the characteristics of cultural routes and the needs of tourists.

(4) To establish route committees or associations for daily management and strategy development. They can be responsible for building the necessary transport and arts infrastructure, organizing regular cultural events, or creating and updating tourist information platforms.

(5) To encourage communities along the route to be involved in tourism development and route planning from the ground up. This strategy not only helps to enhance the

sense of participation and belonging of community members but also promotes the conservation and rational use of cultural heritage.

## 7. Conclusions

This study analyzed the relationship between cultural tourism and cultural routes. Cultural routes can provide excellent travel experiences. Cultural tourism protects cultural corridors and preserves area culture and heritage because it can boost the economy of historical areas, integrate cultural heritage resources, and make them more visible and appealing, thus guiding and promoting people's understanding of cultural heritage to safeguard and inherit it. By reviewing current research on the tourist development of cultural routes, our study identified three categories based on route function, context, and content; three planning patterns; and eight tools for regional conservation and development. Establishing route committees or associations and integrating tourism management are the two most common tools. These tools aim to address the destruction of historic sites, lack of scenic quality, urban sprawl, and depopulation of historic areas, particularly for the poor historical remains of ancient trade or transportation routes.

Nevertheless, it is important to recognize that this research has significant constraints. First, the study primarily emphasizes the theoretical framework and qualitative analysis but lacks actual evidence and data to support the stated results. Second, the study focuses exclusively on cultural routes as defined within a specific historical context, potentially disregarding other forms of cultural tourism experiences and routes that do not strictly adhere to the criteria outlined in this research. This limitation may hinder a more comprehensive understanding of the dynamic connection between cultural tourism and the conservation of cultural heritage. Furthermore, although the study identified tools for the conservation and development of cultural routes, the practical application and effectiveness of these tools in real-world settings were not exhaustively examined. The evaluation of the tools was also more limited to the conclusions of previous researchers.

There are still many problems and challenges in the process of tourism development along cultural routes. The following aspects are proposed for future research: (1) establishment of a planning and management framework from the whole to the local level; (2) creation of assessment tools to achieve the integration of objectives related to heritage preservation, local aesthetics, recreation, and the specific interests of the various stakeholders, including the public, professionals, and decision-makers; and (3) all-encompassing assessment of the cultural routes from the pre-tourism phase, mid-term, and post-tourism phases to balance the preservation and utilization of cultural heritage and avoid the secularization of the symbolic significance of the routes.

**Author Contributions:** Methodology, X.L.; software, X.L.; validation, Z.S., X.T. and X.L.; formal analysis, X.L.; investigation, X.L.; resources, X.L.; data curation, X.L.; writing—original draft preparation, X.L.; writing—review and editing, Z.S., Q.M. and X.T; visualization, X.L.; supervision, Z.S. and Q.M.; project administration, Z.S. All authors have read and agreed to the published version of the manuscript.

**Funding:** This research received no external funding.

**Conflicts of Interest:** The authors declare no conflicts of interest.

## Appendix A

**Table A1.** Study Cases and references.

| No. | Name | Country | Types | Planning Patterns | Tools | References |
|---|---|---|---|---|---|---|
| 1 | Via Regia | France, Germany, Poland, Ukraine, Belarus | Trade, migration, or transportation routes | Point-based planning | Official digital platform | [2,53,77] |
| | | | | | Constructing infrastructure facilities along the routes | |
| | | | | | Establishing route committees or associations | |
| 2 | Rideau Canal | Canada | Heritage canal or valley | Area-based planning | Integral tourism management and route planning | [39,75] |
| | | | | | Community participation | |
| | | | | | Establishing route committees or associations | |
| | | | | | Holding cultural events | |
| | | | | | Offering itinerary planning and travel programs | |
| 3 | The Danube region | Germany, Austria, Hungary, Slovakia, Romania, Bulgaria, Moldova, Serbia, Croatia | Heritage canal or valley | Area-based planning | Community participation | [37,47,78, 79] |
| | | | | | Constructing infrastructure facilities along the routes | |
| | | | | | Integral tourism management and route planning | |
| | | | | | Establishing route committees or associations | |
| 4 | The Way of St. James (the Camino de Santiago) | Spain, France, Portugal | Pilgrimage route | Linear-based planning | Commercialization | [38,51,53, 66,80–82] |
| | | | | | Offering itinerary planning and travel programs | |
| | | | | | Holding cultural events | |
| | | | | | Community participation | |
| | | | | | Constructing infrastructure facilities along the routes | |
| | | | | | Official digital platform | |
| | | | | | Integral tourism management and route planning | |
| | | | | | Establishing route committees or associations | |
| 5 | Via Francigena | England, France, Switzerland, Italy | Pilgrimage route | Linear-based planning | Integral tourism management and route planning | [31,45,55, 56,83,84] |
| | | | | | Establishing route committees or associations | |
| | | | | | Official digital platform | |
| | | | | | Offering itinerary planning and travel programs | |
| | | | | | Community participation | |
| | | | | | Commercialization | |
| | | | | | Holding cultural events | |
| | | | | | Constructing infrastructure facilities along the routes | |
| 6 | The Phoenician Way | Albania, Belgium, Croatia, Cyprus, France, Greece, Italy, Lebanon, Malta, Spain, Tunisia, Slovenia, Ukraine | Trade, migration, or transportation routes | Point-based planning | Establishing route committees or associations | [3,85–87] |
| | | | | | Community participation | |
| | | | | | Holding cultural events | |

**Table A1.** *Cont.*

| No. | Name | Country | Types | Planning Patterns | Tools | References |
|---|---|---|---|---|---|---|
| 7 | The Qhapaq Ñan | Peru, Argentina, Bolivia, Chile, Colombia, Ecuador | Trade, migration, or transportation routes | Point-based planning | Establishing route committees or associations | [48,49,73, 88] |
| | | | | | Community participation | |
| | | | | | Commercialization | |
| 8 | Saint Martin of Tours Route | Austria, Belgium, Croatia, France, Germany, Hungary, Italy, Luxembourg, Netherlands, Slovak, Slovenia, Poland | Pilgrimage route | Point-based planning | Establishing route committees or associations | [46] |
| | | | | | Official digital platform | |
| | | | | | Community participation | |
| | | | | | Holding cultural events | |
| | | | | | Integral tourism management and route planning | |
| | | | | | Commercialization | |
| 9 | Iron Curtain Trail | Austria, Belgium, Croatia, Czech Republic, Germany, Greece, Hungary, Lithuania, Serbia, Slovak Republic, Turkey | Historical borders | Linear-based planning | Official digital platform | [59] |
| | | | | | Constructing infrastructure facilities along the routes | |
| | | | | | Establishing route committees or associations | |
| | | | | | Integral tourism management and route planning | |
| 10 | Via Romea Germanica | Germany, Italy, Austria | Pilgrimage route | Linear-based planning | Official digital platform | [59] |
| | | | | | Integral tourism management and route planning | |
| | | | | | Holding cultural events | |
| | | | | | Establishing route committees or associations | |
| | | | | | Constructing infrastructure facilities along the routes | |
| 11 | Ibar Valley | Serbia | Heritage canal or valley | Area-based planning | Holding cultural events | [61] |
| | | | | | Integral tourism management and route planning | |
| | | | | | Establishing route committees or associations | |
| 12 | The Path of Peace | Italy | Military route | Linear-based planning | Integral tourism management and route planning | [62] |
| | | | | | Offering itinerary planning and travel programs | |
| | | | | | Constructing infrastructure facilities along the routes | |
| | | | | | Establishing route committees or associations | |
| | | | | | Commercialization | |
| 13 | Great Ocean Road | Australia | Military route | Linear-based planning | Constructing infrastructure facilities along the routes | [57] |
| | | | | | Integral tourism management and route planning | |
| | | | | | Establishing route committees or associations | |

**Table A1.** *Cont.*

| No. | Name | Country | Types | Planning Patterns | Tools | References |
|---|---|---|---|---|---|---|
| 14 | Batik Cultural Route | Indonesia | Trade, migration, or transportation routes | Point-based planning | Official digital platform | [68] |
| | | | | | Establishing route committees or associations | |
| 15 | St. Paul Trail | Turkey | Pilgrimage route | Linear-based planning | Official digital platform | [89] |
| | | | | | Integral tourism management and route planning | |
| | | | | | Establishing route committees or associations | |
| 16 | The Lycian Way | Turkey | Trade, migration, or transportation routes | Linear-based planning | Integral tourism management and route planning | [89] |
| | | | | | Establishing route committees or associations | |
| 17 | Tamsui–Kavalan trails | China (Taiwan) | Trade, migration, or transportation routes | Linear-based planning | Integral tourism management and route planning | [50] |
| | | | | | Establishing route committees or associations | |
| | | | | | Community participation | |
| | | | | | Holding cultural events | |
| 18 | The Phrygian Way | Turkey | Trade, migration, or transportation routes | Linear-based planning | Integral tourism management and route planning | [74] |
| | | | | | Establishing route committees or associations | |
| | | | | | Commercialization | |
| 19 | The Kumano Kodo Route | Japan | Pilgrimage route | Linear-based planning | Integral tourism management and route planning | [23] |
| | | | | | Establishing route committees or associations | |
| | | | | | Community participation | |
| | | | | | Holding cultural events | |
| 20 | Oscypek Trail | Poland | Trade, migration, or transportation routes | Area-based planning | Integral tourism management and route planning | [40] |
| | | | | | Establishing route committees or associations | |
| 21 | Roman Emperors and Danube Wine Route | Albania, Bosnia and Herzegovina, Bulgaria, Croatia, Hungary, Montenegro, north Macedonia, Romania, Serbia, Slovenia | Trade, migration, or transportation routes | Point-based planning | Integral tourism management and route planning | [44,72] |
| | | | | | Official digital platform | |
| | | | | | Commercialization | |
| | | | | | Establishing route committees or associations | |
| 22 | The Great Wall in the Ming Dynasty | China | Historical borders | Linear-based planning | Integral tourism management and route planning | [54,90] |
| | | | | | Establishing route committees or associations | |

**Table A1.** *Cont.*

| No. | Name | Country | Types | Planning Patterns | Tools | References |
|---|---|---|---|---|---|---|
| 23 | The Illinois and Michigan Canal | USA | Heritage canal or valley | Area-based planning | Integral tourism management and route planning | [41,76] |
| | | | | | Establishing route committees or associations | |
| | | | | | Community participation | |
| | | | | | Constructing infrastructure facilities along the routes | |
| | | | | | Offering itinerary planning and travel programs | |
| 24 | The Grand Canal | China | Heritage canal or valley | Area-based planning | Integral tourism management and route planning | [52,58,91] |
| | | | | | Establishing route committees or associations | |
| 25 | Meiguan Historical Trail | China | Trade, migration, or transportation routes | Area-based planning | Integral tourism management and route planning | [92] |
| | | | | | Establishing route committees or associations | |
| 26 | Cane River | USA | Heritage canal or valley | Area-based planning | Community participation | [35,36] |
| | | | | | Establishing route committees or associations | |
| | | | | | Integral tourism management and route planning | |
| 27 | Blackstone River Valley | USA | Heritage canal or valley | Area-based planning | Integral tourism management and route planning | [35,36] |
| | | | | | Establishing route committees or associations | |
| 28 | Delaware and Lehigh National Heritage Corridor | USA | Heritage canal or valley | Area-based development | Integral tourism management and route planning | [35,36] |
| | | | | | Establishing route committees or associations | |
| 29 | The Piast Trail | Poland | Trade, migration, or transportation routes | Area-based planning | Establishing route committees or associations | [42] |
| | | | | | Integral tourism management and route planning | |
| 30 | The Underground Military Galleries of the Petrovaradin Fortress | Serbia | Military route | Area-based planning | Integral tourism management and route planning | [43] |
| | | | | | Construction of infrastructure facilities along the routes | |
| | | | | | Establishing route committees or associations | |

**Table A1.** *Cont.*

| No. | Name | Country | Types | Planning Patterns | Tools | References |
|---|---|---|---|---|---|---|
| 31 | The Silk Road | Albania, Armenia, Azerbaijan, Bangladesh, Bulgaria, China, Croatia, Korea, Egypt, Georgia, Greece, Indonesia, Iran, Iraq, Israel, Italy, Japan, Kazakhstan, Kyrgyzstan, Mongolia, Pakistan, Republic of Korea, Romania, Russia, San Marino, Saudi Arabia, Spain, Syria, Tajikistan, Turkey, Turkmenistan, Ukraine, Uzbekistan | Trade, migration, or transportation routes | Point-based planning | Establishing route committees or associations<br>Official digital platform | [25–27] |
| 32 | Erie Canal | USA | Heritage canal or valley | Area-based planning | Constructing infrastructure facilities along the routes<br>Establishing route committees or associations<br>Integral tourism management and route planning<br>Official digital platform | [64,93] |
| 33 | Chishui River | China | Trade, migration, or transportation routes | Point-based planning | Constructing infrastructure facilities along the routes | [60] |
| 34 | Canal du Midi | France | Heritage canal or valley | Area-based planning | Constructing infrastructure facilities along the routes<br>Official digital platform<br>Establishing route committees or associations<br>Integral tourism management and route planning | [53,94] |
| 35 | Cyprus Government Railways | Cyprus | Railway heritage | Point-based planning | Constructing infrastructure facilities along the routes | [65] |
| 36 | Puffing Billy | Austria | Railway heritage | Linear-based planning | Integral tourism management and route planning<br>Establishing route committees or associations<br>Constructing infrastructure facilities along the routes<br>Community participation<br>Holding cultural events<br>Offering itinerary planning and travel programs | [63] |
| 37 | Kuranda Train | Austria | Railway heritage | Linear-based planning | Integral tourism management and route planning<br>Community participation<br>Official digital platform<br>Offering itinerary planning and travel programs | [63] |
| 38 | Route 66 | USA | Highway heritage | Linear-based planning | Integral tourism management and route planning<br>Establishing route committees or associations | [95] |

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
