# Peer review of "Cultural Routes as Cultural Tourism Products for Heritage Conservation and Regional Development: A Systematic Review"

_heritage, doi:10.3390/heritage7050114_

Round 1

Reviewer 1 Report

Comments and Suggestions for Authors

Dear Authors,

the idea of cultural routes is really important in the contemporary heritage studies because it contributes to the logical arrangement of the international cultural and historical heritage and its easier exploitation for the purposes of tourism. The related publications are rather numerous, and it is very reasonable to synthesize the related knowledge. The reviewed manuscript attempts to solve this task. On the one hand, the Authors offer a detailed and original view of the noted idea, and they were able to analyze a lot of published data. Undoubtedly, their work can become a new word in the understanding of cultural heritage. On the other hand, some inaccuracies (also methodological), inconsistencies, and lacking information are visible. The research problem and questions are formulated and argued, and the manuscript is structured and referenced adequately. The illustrations are also optimal. Nonetheless, various improvements are necessary regarding the above-mentioned weaknesses, and I hope my comments will help you to realize what and where should be done.

1)      Abstract (and also below): cultural routes are necessarily ancient? Or may be better historical? For instance, if a route reflects the cultural experience of the 17th century, is it ancient? Moreover, what about the modern culture? – Routes can also be associated to it. Finally, are cultural routes based on only heritage? – What if they are based on the 21st century music? (or this option is impossible?)

2)      Key words: I encourage you to avoid the words already existing in the title.

3)      Subsection 1.1 offers the only European view. But this is the only one of many possible views! What about the other parts of the world and the different countries?

4)      Subsection 1.2: WOT -> WTO. And what about UNESCO WHS?

5)      Subsection 2.1: why Web of Science and Google Scholar, but not Scopus. Please, argue your choice! And why review articles were excluded as you explain? (to me, these are the most precious sources of information!)

6)      Subsection 2.2 and 2.3: what about aesthetics of cultural routes?

7)      Subsection 3.1: USA and China are not regions, but countries, whereas Europe is a too general domain.

8)      Figure 2: America -> USA, regions -> countries, avoid Greenland because this is not a country.

9)      Is your section 4 based on your bibliographical survey? If so, you have to cite the sources in the text more extensively. I see various general statements and examples, and these must be supported by the citations because this is not the original information (the same issue exists also in the subsection 5.1).

10)  Figure 5: is it your own drawing? If so, how is it related to the literature analysis? And why you believe that the Silk Road looked so?  - There are alternative schemes showing its configuration, and you have to indicate whose opinion you follow.

11)  Subsection 5.1: what you measure is NOT the actual usage of the tools, but the relative attention to them by the researchers. Please, explain this and reconsider your statements where necessary.

12)  Although your manuscript is a systematic review, it needs a full-scale section “Discussion” where you have to offer the interpretations of your findings (why the situation is so as registered?), to specify some practical recommendations, and to outline research/knowledge biases. Your findings should also be attached to the international research experience. I also believe that any transdisciplinary inferences are possible.

13)  Conclusions: please, add the limitations of your analysis and the perspectives for future studies.

14)  The writing needs evident polishing regarding the quality of English. I wonder why “cultural routes” are capitalized somewhere. I recommend to avoid too short paragraphs and to be consistent with citing the sources ([x] style is recommended by the journal).

15)  Appendix A: America -> USA.

Comments on the Quality of English Language

The writing needs evident polishing regarding the quality of English. I wonder why “cultural routes” are capitalized somewhere. I recommend to avoid too short paragraphs and to be consistent with citing the sources ([x] style is recommended by the journal).

Author Response

Thank you for the constructive comments from the reviewers on the manuscript titled “Cultural routes as cultural tourism products for heritage conservation and regional development :A systematic review.” We have completed thorough English language editing and made specific content revisions as suggested. These two modifications are highlighted in different colors for clarity.We have described the details of the problem modification in the attachment.

Reviewer 2 Report

Comments and Suggestions for Authors

The article offers an interesting analysis of cultural routes and tourism, through a review of the literature, delving into the research on those planning and management tools most repeated in the 68 articles finally studied. The structure of the paper is well designed and its reading allows us to advance our knowledge of this scientific field, so its publication is recommended, after making some small corrections. In general, it is necessary to reread the text to correct typographical errors, such as the repetition of line 110 on page 3 (is a is a) or the numerous errors in the text in Figure 1. On the other hand, I would like to make a clarification regarding the Camino de Santiago. There is no differentiated path "from Sarria", it is part of the French Way that from this country and through four different routes crosses the Pyrenees and runs through the north of Spain. There are other routes to Santiago de Compostela that are also World Heritage Sites. In total, thousands of kilometers of cultural pilgrimage routes from different places (Portugal, France, United Kingdom, Germany, Italy, Poland, etc.) that converge in Santiago de Compostela. What happens is that to win the jubilee, that is, so that the pilgrimage allows sins to be forgiven for Catholics, at least 100 kilometers must be traveled on foot (or 200 on horseback or bicycle). That is why this final stretch of the French Way (just over 100 kilometers) is highlighted, which would be enough to receive absolution in the Cathedral of Santiago.

Comments on the Quality of English Language

English Language is fine. They have just to make corrections of typos

Author Response

Thank you for the constructive comments from the reviewers on the manuscript titled “Cultural routes as cultural tourism products for heritage conservation and regional development :A systematic review.” We have completed thorough English language editing and made specific content revisions as suggested. These two modifications are highlighted in different colors for clarity.

The  content (especially the case study of the Camino de Santiago), English language and figures have been modified according to the reviewer's comments.

Round 2

Reviewer 1 Report

Comments and Suggestions for Authors

Dear Authors,

thanks for your improvements and responses! I recommend your manuscritp for acceptance.

Comments on the Quality of English Language

The English is better, and I hope the minor linguistic check provided by MDPi for accepted manuscripts together with proof preparation will bring this work fully in order.